# Paracrine Regulation of Alveolar Epithelial Damage and Repair Responses by Human Lung-Resident Mesenchymal Stromal Cells

**DOI:** 10.3390/cells10112860

**Published:** 2021-10-23

**Authors:** Dennis M. L. W. Kruk, Marissa Wisman, Jacobien A. Noordhoek, Mehmet Nizamoglu, Marnix R. Jonker, Harold G. de Bruin, Karla Arevalo Gomez, Nick H. T. ten Hacken, Simon D. Pouwels, Irene H. Heijink

**Affiliations:** 1Medical Center Groningen, Department of Pathology and Medical Biology, University of Groningen, 9713 GZ Groningen, The Netherlands; d.m.l.w.kruk@umcg.nl (D.M.L.W.K.); m.wisman@umcg.nl (M.W.); jacobiennoordhoek@hotmail.com (J.A.N.); m.nizamoglu@umcg.nl (M.N.); m.r.jonker@umcg.nl (M.R.J.); h.g.de.bruin@umcg.nl (H.G.d.B.); k.f.arevalo.gomez@umcg.nl (K.A.G.); s.d.pouwels@umcg.nl (S.D.P.); 2University Medical Center Groningen, Groningen Research Institute for Asthma and COPD, University of Groningen, 9713 GZ Groningen, The Netherlands; nick.ten.hacken@gmail.com; 3University Medical Center Groningen, Department of Pulmonary Diseases, University of Groningen, 9713 GZ Groningen, The Netherlands

**Keywords:** COPD, emphysema, cell therapy, lung repair, MSCs, alveolar epithelium, growth factors, regenerative medicine, organoids

## Abstract

COPD is characterized by irreversible lung tissue damage. We hypothesized that lung-derived mesenchymal stromal cells (LMSCs) reduce alveolar epithelial damage via paracrine processes, and may thus be suitable for cell-based strategies in COPD. We aimed to assess whether COPD-derived LMSCs display abnormalities. LMSCs were isolated from lung tissue of severe COPD patients and non-COPD controls. Effects of LMSC conditioned-medium (CM) on H_2_O_2_-induced, electric field- and scratch-injury were studied in A549 and NCI-H441 epithelial cells. In organoid models, LMSCs were co-cultured with NCI-H441 or primary lung cells. Organoid number, size and expression of alveolar type II markers were assessed. Pre-treatment with LMSC-CM significantly attenuated oxidative stress-induced necrosis and accelerated wound repair in A549. Co-culture with LMSCs supported organoid formation in NCI-H441 and primary epithelial cells, resulting in significantly larger organoids with lower type II-marker positivity in the presence of COPD-derived versus control LMSCs. Similar abnormalities developed in organoids from COPD compared to control-derived lung cells, with significantly larger organoids. Collectively, this indicates that LMSCs’ secretome attenuates alveolar epithelial injury and supports epithelial repair. Additionally, LMSCs promote generation of alveolar organoids, with abnormalities in the supportive effects of COPD-derived LMCS, reflective of impaired regenerative responses of COPD distal lung cells.

## 1. Introduction

Chronic Obstructive Pulmonary Disease (COPD) is a prevalent inflammatory lung disease that has high mortality. The main risk factor for COPD is the inhalation of noxious particles and gasses, such as cigarette smoke and air pollutants, which in combination with genetic susceptibility lead to inflammation, lung tissue damage and aberrant tissue repair in COPD patients. The disease is characterized by mucus hypersecretion (chronic bronchitis), airway wall thickening and/or destruction of the alveoli with airspace enlargement (emphysema) [1], leading to airflow limitation. Currently, there is no cure for the disease and the alveolar damage is irreversible. Therefore, there is an urgent need for novel treatment strategies that halt or reverse the progressive loss of lung function.

Stem-cell-based approaches have shown promising results in pre-clinical models of emphysema [2]. The most widely described stem cell population that has been used in these approaches is the mesenchymal stem/stromal cell (MSC). MSCs have been successfully used in numerous clinical trials for a variety of clinical indications, demonstrating dampened immune reactions and enhanced regeneration [3,4,5,6]. MSCs are multipotent stem cells that can be derived from various stromal tissues, including bone marrow, adipose tissue, umbilical cord and adult human lung [2,7,8], sharing common features such as the expression of mesenchymal surface markers, ability to differentiate into multiple lineages and clonogenic potential [9,10]. The beneficial effects in the treatment of various diseases have been mainly attributed to paracrine mechanisms, secreting regenerative growth factors and immune-modulatory/anti-inflammatory factors, including microRNAs [2]. The use of MSCs has been widely evaluated for improvement of lung performance in animal models of emphysema, confirming reduced inflammation, while supporting repair of alveolar damage and restoring lung structure [8]. Whereas human clinical trials with autologous bone marrow-derived MSCs (BM-MSCs) demonstrated safety and reduced circulating levels of inflammatory marker C-reactive protein (CRP), beneficial effects on lung function in emphysema patients were limited [11,12]. In a more recent study, BM-MSC administration improved lung function in COPD patients with high systemic levels of CRP, although it remains to be established whether these effects can be mainly attributed to anti-inflammatory effects or whether tissue repair was also supported [13]. A pitfall in previous clinical trials may have been the intravenous administration of MSCs [14] and the use of MSCs from other locations than the lung. MSCs from other sources may be less equipped to support lung tissue repair, as tissue-resident lung-derived MSCs (LMSCs) have been reported to possess unique lung-specific properties [15], while BM-MSCs were reported to lack specific mechanisms that enhance their lung retention [16].

Another challenge may be that MSCs from COPD patients display intrinsic defects due to (epi)genetic factors, including expression of COPD susceptibility genes. Thus, the use of autologous cells may limit the effectiveness of cell-based strategies. Of note, we recently observed that LMSCs from severe COPD patients express lower levels of hepatocyte growth factor (HGF) and fibroblast growth factor (FGF)10 [10], which both play critical roles in distal lung repair processes [17,18,19,20], compared to those obtained from non-COPD lung tissue [10]. This suggests that abnormalities in LMSC function may contribute to the failing lung tissue repair process in COPD, although it is currently unknown whether the observed differences between endogenous COPD and non-COPD-derived LMSCs have consequences for the protection against epithelial damage and/or the support of epithelial repair. Moreover, effects of LMSCs on alveolar epithelial damage and repair responses have not been well characterized.

Before LMSCs should be considered for therapeutic strategies in emphysema, it is important to gain more insight into the role of endogenous lung-resident MSCs in alveolar epithelial injury and repair responses and potential defects in the support of COPD-derived LMSCs. 

We hypothesized that LMSCs derived from human lung tissue are able to reduce alveolar epithelial damage and promote alveolar epithelial repair and regenerative responses in a paracrine manner, and that these effects are compromised in COPD-derived LMSCs. To assess this, we isolated MSCs from lung tissue of COPD and non-COPD donors and investigated effects of their conditioned medium on oxidative stress-induced cellular damage, repair upon electric field-injury and scratch-wounding in the human lung cells lines A549 and NCI-H441. In addition, we studied regenerative responses using LMSCs in co-culture with lung epithelial cells in a previously published organoid model [21], recapitulating critical processes during distal lung regeneration. To increase the relevance of the findings, we also assessed the organoid forming ability of endogenous lung cells from COPD and non-COPD control donors.

## 2. Materials and Methods

### 2.1. Subjects 

Distal lung tissue was derived from a total of 12 emphysema patients with GOLD stage III–IV COPD undergoing lung transplantation or lung volume reduction surgery and from leftover lung material of 12 non-COPD controls undergoing tumor resection surgery. See Table 1 for patient characteristics. Tissue was collected as far distant from the tumor as possible, checked for abnormalities by an experienced pathologist and if indicated, the tissue was excluded from our study. The study protocol was consistent with the Research Code of the University Medical Center Groningen (https://www.rug.nl/umcg/research/documents/research-code-info-umcg-nl.pdf (accessed on 17 October 2021)) and national ethical and professional guidelines (“Code of conduct; Dutch federation of biomedical scientific societies”, htttp://www.federa.org (accessed on 17 October 2021)).

### 2.2. LMSC Isolation and Culture

LMSCs were acquired from peripheral parenchymal lung tissue by culture from tissue explants in 1:1 Dulbecco’s Modified Eagle’s medium DMEM/Ham’s F-12 medium (Gibco, Waltham, MA, USA) on fibronectin-coated plates as described previously [10]. In short, tissue fragments were covered in 10 mL/0.25% trypsin-EDTA (Gibco) and incubated at 37 °C for 15 min. Trypsin was neutralized using fetal calf serum (FCS; Hyclone, Logan, UT, USA) and 2 tissue fragments/well were transferred to 6-well plates coated with 30 µg/mL fibronectin and 10 µg/mL BSA (Sigma-Aldrich, Saint Louis, MO, USA). Tissue pieces were covered with 1 mL DMEM/Ham’s F12 medium containing 1% Glutamax (Gibco), 10% FCS, and 100 U/mL penicillin/100 µg/mL streptomycin (Invitrogen, Breda, The Netherlands) per well. Cells were grown to ~30% confluence in 2–3 weeks. Tissue fragments were removed and cells were passaged to uncoated plates. After reaching ~90% confluence, cells were stored in liquid nitrogen. For experiments, cells in passage 2 were seeded in 6-well plates. Cells were grown for 2–3 days to ~90% confluence, serum-deprived overnight and placed into fresh serum-free DMEM/Ham’s F12 medium for 24 hrs. Cell-free supernatants were stored for conditioned medium (CM) experiments with epithelial cells.

### 2.3. Primary Human Lung Cell Isolation and Epithelial Cell Culture

Lung tissue was cut into small pieces (1 mm^3^) and treated overnight at 4 °C with Trypsin/EDTA (0.25%; Gibco), penicillin/streptomycin (100 U/mL; 100 µg/mL), 2 mg/mL Collagenase A (Roche, Basel, Switzerland) and 0.04 mg/mL DNase (Sigma-Aldrich). Next, the suspension was diluted in DMEM containing 0.04 mg/mL DNase and penicillin/streptomycin, homogenized and filtered over two layers of gause filter (Sefar Nitex, Heiden, Switzerland). After centrifugation (500× *g* for 10 min) the pellet was resuspended in lysis buffer (15.5 mM NH_4_CL, 1 mM KHCO_3_, 0.01mM EDTA, 0.04 mg/mL DNase) for 10 min at 4 °C. The remaining pellet was resuspended in Small Airway Epithelial Cell Growth (SAGM) medium (PromoCell, Heidelberg, Germany) supplemented with penicillin/streptomycin, and kept on ice until use within 1–4 hrs. Cell suspensions were seeded into organoid cultures directly or EpCAM^+^ cells were isolated by anti-CD45 (#130-045-801, Miltenyi Biotec, Auburn, AL, USA) and anti-CD31 (#130-091-935, Miltenyi Biotec) coupled-microbeads and passed through LS columns (Miltenyi Biotec) according to the manufacturer’s guidelines. The flow-through was incubated with anti-EpCAM (CD326) microbeads (#130-061-101, Miltenyi Biotec) for positive selection with LS columns. Cells were resuspended in SAGM and kept on ice until use within 1–2 h.

### 2.4. Culture of Human Lung Epithelial Cell Lines 

The human alveolar carcinoma cell line A549 (ATCC, Manassas, VA, USA) was cultured at 37 °C/5% CO2 in RPMI supplemented 10% FCS and penicillin/streptomycin (100 U/mL; 100 µg/mL) in uncoated T25 flasks. The human lung adenocarcinoma cell line NCI-H441 (ATCC) were cultured in DMEM supplemented with Corning^®^ Insulin/Transferrin/Selenium (ITS) Premix Universal Culture Supplement (Corning, Bedford, MA, USA) 10% FCS and penicillin/streptomycin (100 U/mL; 100 µg/mL) in uncoated T25 flasks. Cells were passaged at ~90% confluence using trypsin-EDTA.

### 2.5. Treatment of Epithelial Cell Lines

Cells were seeded in duplicates at a density of 50 × 10^3^ cells in uncoated 24-well plates or at a density of 75 × 10^3^ cells in ECIS arrays and grown for 2–3 days to ~90% confluence. Cells were serum-deprived and incubated with the respective culture media (RMPI or DMEM) and control serum-free RPMI or DMEM/Ham’s F12 medium (negative control) or LMSC-CM in a 1:1 ratio. In the wounding assays, 1% FCS was added to the 1:1 medium mixture as positive control. To assess cell viability and apoptosis/necrosis, cells were treated with 50 µM, 500 µM and 5 mM H_2_O_2_ (Merck) for 4 h and harvested upon trypsin/EDTA treatment for live/dead staining. Wounding by electric field application or scratching were performed as described below.

### 2.6. Cell Viability by Annexin V/PI Staining

The percentage of viable, apoptotic and necrotic cells was determined using annexin-V/PI staining for flow cytometry. Cells were trypsinized and washed twice using cell staining buffer (BioLegend, San Diego, CA, USA) and stained in Annexin-V binding buffer (BioLegend) using 2.5 μL of Annexin-V-FITC (Immunotools, Friesoythe, Germany) and 2.5 μg/mL propidium iodide (Sigma-Aldrich). The percentage of necrotic, (early) apoptotic, and viable cells was measured afterwards using the BD FACS-Calibur (BD Biosciences, Franklin Jakes, NJ, USA) flow-cytometer and data were analyzed using Winlist software (Verity Software House, Topsham, ME, USA). 

### 2.7. Electrical Resistance Measurements and Electric Field Injury

Cells were seeded into electric cell-surface impedance sensing (ECIS) arrays and monitored by the ECIS system (Applied BioPhysics inc., Troy, MI, USA). Resistance was monitored in real-time at a frequency of 400 Hz and capacitance at a frequency of 40 kHz. After 2–3 days, when a confluent monolayer was established and resistance values stabilized, cells were serum-deprived overnight in the presence and absence of LMSC-CM. Cells were wounded by electroporation (30 s, using voltage pulses of 5 V and a frequency of 40 kHz), resulting in a reproducible wound restricted to a small area of the 0.49 mm^2^ electrode [22]. After wounding, cells were monitored for another 24 h to assess the epithelial repair response. 

### 2.8. Scratch Wounding

A scratch was applied to epithelial monolayers using a P100 disposable pipette tip, gently creating a single diagonal scratch across the whole well. Afterwards the well was gently washed, and bright field microscopy images of the complete wounded area were taken directly after scratching (T0) and after 24, and 48 h. Relative wound closure was assessed by measuring the average length from one side of the scratch to another at T0 by derived from 3 measurements using ImageJ. Drawing a line representing the average length at T0 and copying this to the photos of T24-72, to measure the average (3 measurements) length of the scratch relative to the copied line. 

### 2.9. qPCR

After 24 h of incubation with LMSC-CM or negative control medium, A549 cells were harvested for qPCR analysis. Cells were lysed in TRI reagent (MRC, Cincinnati, OH, USA) for RNA isolation using the chloroform extraction method. cDNA synthesis (iScript cDNA synthesis kit (BioRad, Hercules, CA, USA) and qPCR analysis using TaqMan (Life Technologies, Waltham, MA, USA) were performed in accordance to the manufacturer’s instructions. Validated TaqMan probes were used for the assessment of expression of the housekeeping gene *B2M* and *PPIA* and the epithelial growth factors *EGF*, *VEGF*, *WNT5A*, *TGFB* and *IGF1* in technical duplicates.

### 2.10. MRC-5 and LMSC Culture for Organoids

MRC-5 human lung fibroblasts (ATCC) or LMSCs were defrosted and cultured overnight in uncoated flasks/well plates in Ham’s F12 (MRC-5) or DMEM/Ham’s F12 (LMSCs) medium supplemented with 10% FCS, penicillin/streptomycin (100 U/mL; 100 µg/mL) and 1% Glutamax. Cells were proliferation-inactivated with 0.01mg/mL Mitomycin C (Sigma) in their respective media for 2 h, extensively washed, trypsinized, resuspended in RPMI/10% FCS/0.1% Corning^®^ ITS Premix Universal Culture Supplement for co-culture with NCI-H441 in SAGM for co-culture with primary lung cells and kept on ice for 1–2 hrs. 

### 2.11. Organoid Culture of NCI-H441 and Primary Distal Lung Cells

NCI-H441cells (1.25 × 10^3^), EpCAM^+^ primary human lung epithelial cells (5 × 10^3^) cells or unfractionated human lung suspensions (10 × 10^3^ cells) were mixed with an equal number of mitomycin-treated MRC-5 fibroblasts or LMSCs and seeded into 100 µL growth factor-reduced Matrigel (Corning) diluted 2:1 with SAGM supplemented with 1% FCS, seeded in triplicates in 6.5 mm Transwell inserts (Corning, 0.4 µm pore size) and allowed to gel for ~30 min before adding 500 µL SAGM/1% FCS only to the basolateral part of the well [20]. Basolateral medium was replaced three times a week and cultures were maintained at 37 °C in a humidified incubator with 5% CO2.

### 2.12. Quantification of Organoid Size and Number

At day 7 and day 14 after seeding, total numbers of organoids in all 3 inserts were quantified manually using a light microscope at 2× magnification. Diameters were assessed by Nikon Eclipse Ti software (Nikon Instruments Europe, Amsterdam, The Netherlands). Spheres of a diameter >50 µm were defined as organoid and counted [21].

### 2.13. Staining for Type II Pneumocyte Markers SPC and HTII-280

At day 7 and 14 after seeding, inserts were fixed with ice-cold Acetone/Methanol (1:1; Merck) for 20 min at −20 °C or with 4% paraformaldehyde (PFA, Merck) in PBS for 30 min at room temperature. The inserts were stored in 1% BSA/PBS at 4 °C. To detect pro-surfactant C (SPC), inserts were blocked with 0.075% H_2_O_2_ for 30 min at room temperature, incubated with 1:200 primary antibody (anti-SPC antibody, Sigma-Aldrich) or isotype control in PBS/1%BSA/0.1%Triton overnight at 4 °C, washed, incubated with the secondary antibody (1:200 Rabbit anti-Mouse-peroxidase (Dako, Santa Clara, CA, USA) in 1 × PBS/1%BSA/0.1%Triton) followed by the tertiary antibody (1:200 Goat anti-Rabbit-peroxidase (Dako) in 1 × PBS/1%BSA/0.1%Triton) for 30 min. Peroxidase was catalyzed using aminoethyl carbazole (AEC) in 50 mM acetate buffer substrate and Hematoxylin counterstaining of the nuclei. The membranes were mounted on glass slides and the organoids were visualized at 10× with bright field microscopy. To detect HTII-280, inserts were incubated with 1:200 primary antibody (anti- HTII-280; Terrace Biotec, San Francisco, CA, USA) solution in PBS/1%BSA/0.1%Triton, washed overnight at 4 °C and incubated with the secondary antibody solution (1:200 Donkey anti-Mouse-AlexaFluor 647, Life Technologies, Bleiswijk, The Netherlands in 1 × PBS/1%BSA/0.1%Triton) overnight at 4 °C. After washing, DAPI (Sigma-Aldrich; 1:1000)/1 × PBS was added and incubated for 30 min to stain the nuclei, before visualizing the organoids on the Celldiscoverer 7, Zeiss). The 24-wells format inserts were placed on the bottom of a 12-wells plate so the working distance of the microscope could visualize the organoids in the matrigel on the inserts.

### 2.14. Statistics

The Mann–Whitney U test was used when testing for differences between two groups and the Wilcoxon signed rank test was used for paired comparisons between conditions within groups. The Friedman test with Dunn’s correction was used when conditions were compared. Two-way ANOVA was used to compare time elapse curves in the wounding assays. *p* < 0.05 was considered statistically significant.

## 3. Results

### 3.1. Conditioned-Medium from LMSCs Protects from Oxidative Stress-Induced Cell Death

LMSCs were assessed for their protective effects on epithelial damage and repair. First, the effects on oxidative stress-induced alveolar epithelial damage were assessed. A549 cells were exposed to 0, 50 µM, 500 µM and 5 mM H_2_O_2_ for 4 h to induce cellular stress, resulting in increased apoptosis as well as necrosis and a concomitant reduction in viable cells at the highest concentration (Figure 1A–D). Overnight pre-treatment of epithelial cells with LMSC-conditioned medium (CM) significantly reduced the number of necrotic cells and increased the number of viable cells (Figure 1B,C). No significant differences were observed in the ability to protect from oxidative stress between COPD and non-COPD control-derived LMSCs (Figure 1B,C).

### 3.2. Conditioned-Medium from LMSCs Improves Epithelial Repair Migratory and Proliferative Responses upon Injury

Next, we assessed the ability of LMSCs to improve epithelial repair upon injury. Electric field-induced cell death resulted in an almost complete detachment of the cells from the electrode, as described previously [22]. This was reflected by a robust decrease in epithelial low-frequency resistance (Figure 2A,C) and concomitant increase in high-frequency capacitance levels, from which the cells recovered within 0–6 h upon wounding (Figure 2B,C), repopulating the denuded area. After 6 h, the resistance gradually increased further (Figure 2A,C), reflecting the re-establishment of intercellular contacts [22]. LMSC-derived CM significantly affected the epithelial repair response upon wounding, accelerating the initial recovery phase, but attenuating the gradual increase in resistance observed after 6 h upon wounding (Figure 2C). No significant difference was observed between CM obtained from COPD and non-COPD-derived LMSCs (Figure 2D).

In addition to electric field-induced injury, the scratch-wounding model was used, creating a larger area of damage where cells not only need to migrate, but also a need to proliferate to recover the monolayer (Figure 3A). A549 cells repopulated ~85% of the damaged area within 48 h in the presence of 1% FCS. In the absence of FCS, A549 cells repopulated only ~25% of the wounded area (Figure 3B), indicating that FCS is required for epithelial proliferative responses. LMSC-CM improved recovery of the monolayer, significantly accelerating the repopulation of the wounded area from ~25% to ~50% within 48 h (Figure 3B). Although the effect of CM from COPD-derived LMSCs was slightly less beneficial, the difference between the groups was not significant. NCI-441 cells have been described to reflect the primary human distal epithelium closely, e.g., with respect to expression of transporter genes [23]. Upon scratch wounding of NCI-H441 monolayers, the cells also largely repopulated the wounded area within 48 h, however it was difficult to accurately quantify the recovery of the monolayer (see online data Appendix A for a representative image). We did not further determine effects of LMSC-CM here. 

To gain further insight into the epithelial changes induced by LMSC paracrine factors, we assessed the expression of several epithelial growth factors in A549 cells upon incubation with LMSC-CM. While 48 h of treatment with LMSC-CM did not affect epithelial expression of epidermal growth factor (EGF) and vascular endothelial growth factor (VEGF), it reduced the mRNA expression of transforming growth factor (TGF)-β, and increased mRNA expression of WNT-5A (Figure 4). Insulin-like growth factor (IGF)-1 was not detectable. Again, no significant differences were observed in A549 responses to CM from COPD or control-derived LMSCs (Figure 4).

### 3.3. Regenerative Paracrine Effects of LMSCs in an Organoid Model

Next, we assessed the ability of LMSCs to support the formation of organoids, where alveolar epithelial cells assemble into self-organized lung epithelial structures as a model of regenerative responses [24], for which stromal support is indispensable [21]. Using A549 cells in this model with mitomycin-treated MRC-5 cells for stromal support, we observed that high numbers of spheroids formed, but due to their irregular shape and clustering, spheroid size and numbers were difficult to quantify (Online data Appendix A). Replacing A549 by NCI-H441 cells, regular round-shaped spheroids developed, which significantly increased in size from day 7 to day 14 (Figure 5A), but not in number (data not shown). When replacing MRC-5 cells by (mitomycin-treated) LMSCs, we observed comparable organoid forming capacity (data not shown), although MRC-5-supported organoids were larger at day 7 (Figure 5A). Notably, when comparing the support of COPD and non-COPD control-derived LMSCs, the number of large-sized organoids (100–200 µm diameter) formed by NCI-H441 cells was significantly higher in the presence of COPD-derived LMSCs. This difference was significant at day 7, but no longer at day 14 (Figure 5B). Visualization of alveolar type II cells by SPC staining confirmed that NCI-H441 cells were able to generate spheroids with alveolar characteristics. The increase in organoid size in the presence of COPD-derived LMSCs was accompanied by lower SPC positivity (Figure 5C), especially at day 7, while at day 14 all organoids became SPC positive.

### 3.4. COPD-Derived Lung Cells Form Larger Alveolospheres Than Control-Derived Lung Cells 

To enhance the relevance of our findings with respect to the regenerative effects of COPD- and non-COPD-derived LMSCs, the organoid forming potential of COPD-derived epithelial progenitors from distal lung tissue was studied using the design as depicted in Figure 6A. EpCAM^+^ sorted cells were able to form spheres when using MRC-5 cells as supporting cells (Figure 6B,C). Upon replacing MRC-5 cells by a pooled fraction of 3 control-derived mitomycin-treated LMSC cultures, we observed that LMSCs were able to support the formation of organoids to a similar extent as MRC-5 cells, resulting in similar numbers with comparable size (Figure 6B,C). Next, we compared organoid formation of EpCAM^+^ cells from COPD and non-COPD donors using a stable batch of stromal cells (MRC-5). The size (data not shown) as well as colony forming efficiency of EpCAM^+^ cells from COPD and non-COPD lungs were similar (Figure 6D). We also studied the generation of alveolospheres from unfractionated cellular lung tissue suspensions, containing a lower percentage of epithelial (pan-cytokeratin^+^) cells and other cell types being present, including CD90/CD29^+^ stromal cells, CD68^+^ macrophages and CD31^+^ endothelial cells (Online data Appendix A). Notably, these fractions were able to form organoids without additional stromal support (Figure 6C) and in the presence of MRC-5 cells had higher spheroid forming potential compared to EpCAM^+^ cultures cells, correcting for input of epithelial (pan-cytokeratin^+^) cells (Figure 6D). Staining for the type II markers SPC and HTII-280 indicated that epithelial progenitors from human distal lung tissue were able to generate spheroids with alveolar characteristics (Figure 6E,F). While the numbers of organoids from COPD-derived suspensions were not different compared to control-derived suspensions, organoid forming capacity continued to increase until day 14 (Figure 6G). Moreover, similar to the larger organoids generated by NCI-H441 cells in the presence of COPD-derived LMSCs (Figure 5B), COPD-derived suspensions generated larger organoids, with a significant difference between COPD and control at day 14 (Figure 6H). 

## 4. Discussion

In the current study, we investigated whether endogenous LMSCs exert beneficial effects on distal lung epithelial damage and repair responses within the context of COPD. We observed that LMSCs are able to reduce damage in response to oxidative stress and to promote repair in response to different types of injury in a paracrine manner. Additionally, LMSCs supported alveolar epithelial regenerative responses in an organoid model. Pretreatment of the alveolar epithelial cell line A549 with conditioned-medium from LMSCs protected against H_2_O_2_-induced cell death, without a significant difference in the effect of LMSCs from COPD and non-COPD lungs. When applying electric field-induced injury or scratch-wounding, pre-treatment with soluble factors from LMSCs resulted in accelerated recovery of the alveolar epithelial monolayer, improving migratory and proliferative responses, which was accompanied by increased gene expression of *WNT5A* and decreased expression of *TGFB*. LMSC-derived condition medium impaired the establishment of cell-cell contacts as measured by electrical resistance [22], potentially leading to reduced contact inhibition of migratory and proliferative epithelial responses. Again, we did not observe significant differences in the ability of COPD and non-COPD-derived LMSCs to promote epithelial wound repair responses. Finally, when comparing the ability of COPD and non-COPD-control derived LMSCs to support the generation of alveolospheres in an organoid model, we observed abnormalities in co-cultures with COPD-derived LMSCs. Here, larger organoids developed compared to organoids formed in the presence of control LMSCs. Importantly, similar abnormalities were observed in organoids formed by primary distal lung suspensions from COPD patients, indicating a defect that may be reflective of failing repair responses in COPD lungs. 

To the best of our knowledge, we are the first to demonstrate that human lung-derived MSCs reduce human epithelial cell damage and promote alveolar epithelial regenerative responses. In a murine setting, it was shown that LMSCs promote the self-organizing capacity of alveolar epithelial progenitors in organoids, in which LMSCs of aged mice were impaired [25]. Murine BM-MSCs and their conditioned media are able to support distal lung epithelium-derived organoid formation and increase alveolar differentiation [26]. In A549 cells, human BM-MSCs promoted gap closure upon scratch wounding [27]. Paracrine growth factors implicated in this effect include insulin like growth factor binding protein 7 (IGFBP-7), periostin [27] and KGF [28], the latter two factors which we have observed to be expressed by LMSCs as well. Kennelly et al. showed anti-apoptotic effects of BM-MSCs on alveolar epithelial cells in a mouse model of COPD, which were at least in part mediated by HGF [17]. Anti-apoptotic effects of HGF have also been observed upon cigarette smoke exposure in human bronchial epithelial cells [29]. Nita et al. found that neutralization of HGF attenuated the protective effects of BM-MSC-conditioned medium on alveolar epithelial wound healing upon exposure to cellular stressors and reduced the increase in synthesis of SPC in primary alveolar epithelial cells [30]. SPC is involved in the maintenance of structural integrity, reducing surface tension in the alveoli, and is also a marker of type II alveolar progenitor cells. The secretion of HGF may be thus one of the ways by which LMSCs protect epithelial cells from damage and/or support alveolar epithelial regenerative responses. We previously observed that substantial levels of HGF are secreted by non-diseased LMSCs, which was significantly lower in COPD-derived LMSCs [10]. In line, lower levels of HGF were observed in peripheral epithelial lining fluid of COPD patients compared to non-COPD controls [31], and HGF expression was shown to positively correlate to physical lung parameters such as alveolar diffusion capacity.

We did not find differences in effects on epithelial damage and repair responses between COPD and control-derived LMSCs. This suggests that LMSCs from COPD patients still secrete sufficiently high levels of HGF or additional factors to protect alveolar epithelial cells from oxidative stress-induced damage and to promote epithelial migration and proliferation upon injury. Indeed, the secretome of MSCs contains a plethora of factors that have been implicated in cytoprotective and/or proliferative effects [32], including periostin, KGF, VEGF and IGF-I, which we previously observed to be expressed by LMSCs, at a similar level between COPD and control [10]. IGF-I has been shown to activate Wnt5a in primary rat alveolar epithelial cells, which promoted both wound healing responses and differentiation of type IIs into type I-like cells [33]. On the other hand, Wnt5a signaling has been shown to maintain stemness of murine type II cells [34], while recombinant WNT-5A repressed the differentiation of alveolar progenitors in murine and human organoids [35]. We observed increased expression of *WNT5A* in conjunction with decreased expression of *TGFB* in A549 cells upon exposure to LMSCs’ secretome. TGF-β, in turn, has been shown to impair the ability of stromal cells to support epithelial repair and lung organoid formation [36]. Thus, LMSC-induced alterations in the expression of WNT-5A and TGF-β may contribute to the supportive effects on epithelial repair and regeneration. Again, these effects were not different between LMSCs from COPD and control donors, thus intrinsic defects in other cell types or defects in cellular interactions are more likely to contribute to abnormalities in tissue damage and repair responses in COPD. Further, we cannot exclude the possibility that abnormalities in LMSC responses are only observed in a COPD environment, e.g., in a pro-inflammatory milieu, in the presence of oxidative stress or upon crosstalk within the local ECM in COPD. In this respect, inflammatory cytokines have been shown to increase the capacity of human BM-MSCs to support epithelial wound closure [37]. 

On the other hand, despite the lack of differences between COPD and control in the responses described above, we did observe abnormalities in the support of epithelial regenerative responses by COPD-derived LMSCs. This was assessed by the use of organoid models, which reflect critical processes of lung regeneration [24]. Following injury in lung tissue, alveolar epithelial progenitors spread, migrate, self-renew, proliferate and finally differentiate into alveolar type II and subsequently type I cells to cover the denuded surface and restore the epithelial barrier. The assembly into organized structures requires self-renewal, proliferation and differentiation of epithelial progenitor cells. We observed that LMSCs are able to support the ability to generate alveolospheres. The human lung cell line NCI-H441 formed more large organoids in the presence of COPD-derived LMSCs, with less positivity for type II marker SPC in earlier stages. Organoids derived from COPD lung cell suspension displayed similar abnormalities, forming larger organoids. These differences between COPD and control-derived cultures were not observed when organoids were generated from EpCAM^+^ sorted epithelial cells, indicating that there are no intrinsic defects in COPD-derived alveolar progenitors with respect to the formation of alveolospheres. Rather, communication with other cell types present in these suspensions, including CD90^+^ stromal cells, may be dysregulated. The exact mechanisms of the observed differences between COPD and control require further investigation, as well as to what extent the formation of larger organoids is related to airspace enlargement in COPD. Previously, when studying the organoid forming potential of lung epithelial progenitors in a murine setting, we observed that larger organoids with lower SPC positivity develop in the presence of COPD-related alarmins [38]. We proposed that this may reflect impaired self-renewal of alveolar progenitors or impaired differentiation towards type II pneumocytes. Similarly, the secretome of LMSCs from COPD patients may hamper self-renewal of alveolar progenitors or differentiation of uncommitted progenitors towards the alveolar fate, while instead promoting cell division of progenitors during organoid formation. The previously observed lower expression of HGF and FGF10 in COPD-derived LMSCs compared to those derived from control lungs may be involved [10]. HGF has been shown to play an important supportive role in the repair of alveolar epithelial cells, increasing alveolar epithelial DNA synthesis in a mouse model of acute lung injury [39] and alveolar epithelial organoid formation in vitro [18]. Additionally, murine studies have shown that FGF10 has a critical role in lung developmental and regenerative processes [19,20]. *Fgf10* knock-out mice did not develop lungs [40], while reduced levels of *Fgf10* upon partial knock-out resulted in severe lung hypoplasia, enlargement in the distal lungs and a specific reduction in epithelial cells expressing type II marker surfactant B [20]. In a murine organoid model, the presence of FGF10 promoted branching and differentiation of into particularly distal lung epithelial cells [41]. Thus, lower secretion of both HGF and FGF10 could contribute to the abnormalities observed in LMSC-supported organoid formation, while leaving cytoprotective effects and support of wound healing responses by COPD-derived LMSCs intact. Alternatively, since epithelial cells were co-cultured with LMSCs in the organoid model instead of adding their secreted factors, abnormalities may exist in direct interactions between epithelial cells and LMSCs from COPD patients. The nature of such direct interactions remains unknown. Nonetheless, the observed abnormalities may have important consequences, contributing to impaired alveolar tissue regeneration in COPD. Interestingly, defects in the potential to differentiate have been observed in airway epithelial progenitors from COPD patients using in vitro models [42]. Our current data support the notion of a defective stromal niche in COPD, including LMSCs, which may lead to hampered alveolar epithelial differentiation.

Our study has some limitations. Because of the rapid loss of alveolar cell markers in primary epithelium during 2D culture and in order to keep the epithelial component stable, we used the carcinoma cells lines NCI-H441 and A549 to model epithelial damage and wound repair responses. The limited availability of primary alveolar epithelial cells and inability to form organoids after storage in liquid nitrogen prompted us to use used these cell lines in the organoid assays as well. To enhance the translational value, we compared organoid formation of primary lung cell suspensions from COPD patients and controls. We were not able to compare supportive effects of COPD- and control-derived LMSCs in primary organoid models, as we did not obtain sufficient cell numbers from fresh lung tissue to compare the effects of the different LMSC groups in sufficiently high sample size. Thus, we used a stable batch of stromal (MRC-5) cells, and observed abnormalities in organoid formation by COPD-derived epithelial progenitors when supportive cell populations were present. Together, using this approach, we were able to show that lung-resident MSCs play a critical role in the attenuation of alveolar epithelial damage, in which COPD-derived LMSCs do not display persistent, intrinsic defects upon in vitro expansion in 2D culture. LMSCs also support alveolar epithelial regeneration, in which COPD-derived LMSCs do display intrinsic abnormalities.

## 5. Conclusions

Collectively, we show that paracrine factors from LMSCs reduce distal lung epithelial damage and support repair and regenerative responses. LMSCs from COPD lungs show specific abnormalities in the support of alveolar regenerative responses as demonstrated by the formation of larger organoids. Larger organoids were also generated from lung cell suspensions of COPD patients versus non-COPD controls, and may reflect impaired differentiation into type II cells. These abnormalities should be taken into account when considering autologous LMSCs for cell-based therapeutic strategies in COPD. Moreover, our study identifies endogenous LMSCs as target for therapeutic strategies, contributing to abnormal lung tissue repair in COPD.

## Figures and Tables

**Figure 1 cells-10-02860-f001:**
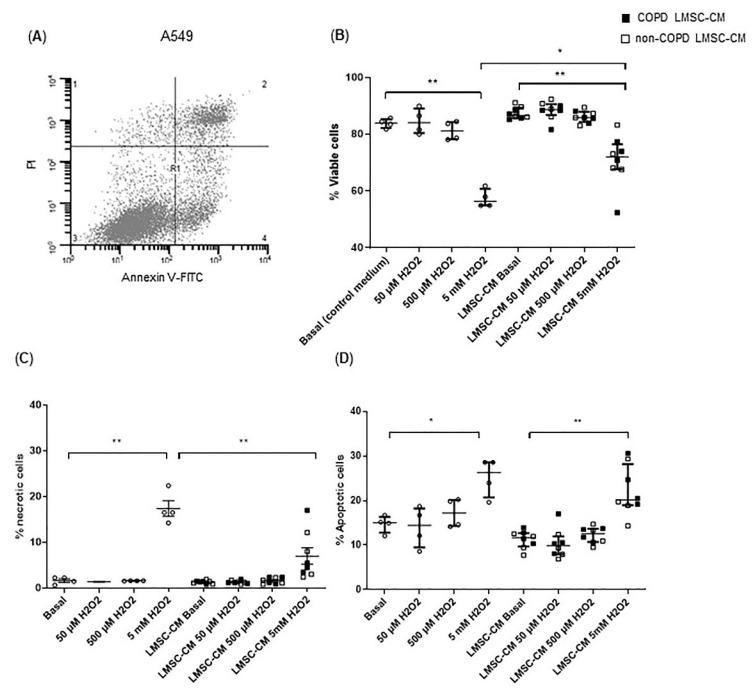
Conditioned medium from lung-resident MSCs (LMSCs) reduces oxidative stress-induced necrotic epithelial cell death. A549 cells were cultured to ~90% confluence and serum-deprived overnight in the presence and absence of conditioned medium (CM) of LMSCs from non-COPD control and COPD donors and treated with 0, 50 µM, 500 µM and 5 mM H_2_O_2_ for 4 h. The percentage of viable, early apoptotic, late apoptotic/necroptotic and necrotic cells was determined using annexin-V/propidium iodide (PI) staining using flow cytometry. (**A**) Representative flow cytometry plot representing the following populations in the respective quadrants: 1. Necrotic cells; 2. Late apoptotic/necroptotic cells; 3. Viable cells; 4. Early apoptotic cells. (**B**) The percentages of viable cells. (**C**,**D**) The percentages of early apoptotic cells. The percentages of necrotic cells was determined using annexin-V/PI staining for flow cytometry. Medians ± interquartile range (IQR) are shown. To test for differences between control medium and LMSC-condition medium, the Friedman test was used. * = *p* < 0.05 and ** = *p* < 0.01 between the indicated values. LMSC-CM from non-COPD donors is indicated by open symbols, from COPD donors by closed symbols. Circles indicate control condition without CM.

**Figure 2 cells-10-02860-f002:**
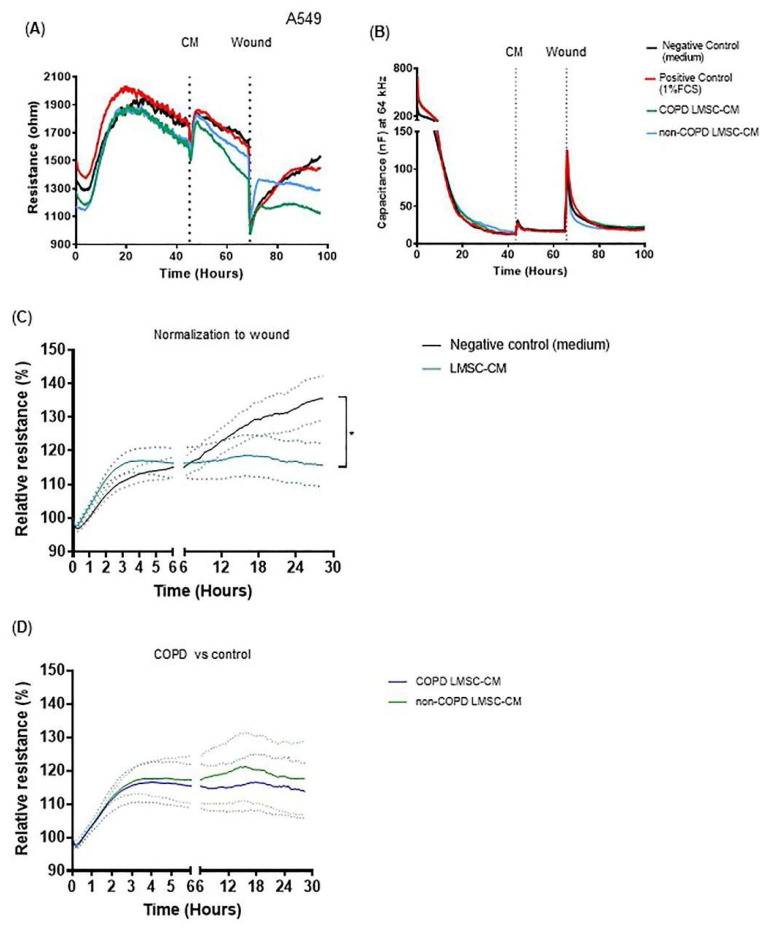
Conditioned medium from lung-resident MSCs (LMSCs) alters recovery of the A549 epithelial cell monolayer after wounding by electroporation. A549 cells were seeded in duplicates in ECIS arrays and grown to confluence for 48 h. Medium was replaced by control medium (negative control), medium containing 1% FCS (positive control) or conditioned medium (CM) from LMSCs of non-COPD control and COPD donors. After 24 h, cells were wounded by electroporation. (**A**) Resistance was measured at a frequency of 400 Hz; a representative plot is shown. (**B**) Capacitance was measured at a frequency of 40 kHz, a representative plot is shown. (**C**) Resistance levels were normalized to the values immediately after wounding in the absence and presence of LMSC-CM from a non-COPD or COPD donor in 4 independent experiments. Mean ± SEM levels are shown (*n* = 8, non-COPD and COPD LMSC-CM combined). (**D**) Resistance levels were normalized to the values immediately after wounding in the presence of LMSC-CM from non-COPD and COPD donors. Mean ± SEM levels are shown (*n* = 4/group). To test for differences between control medium and LMSC-condition medium, 2-way ANOVA was used. * = *p* <0.05 between the indicated values measured over the last 6 h.

**Figure 3 cells-10-02860-f003:**
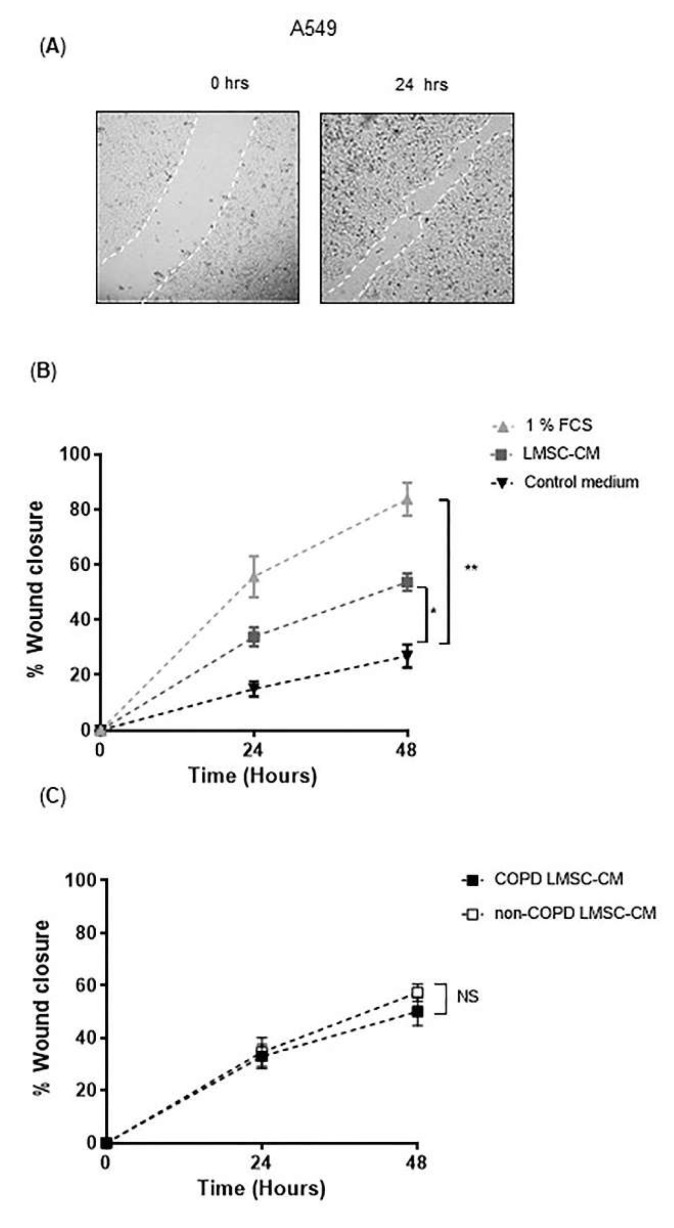
Conditioned medium from lung-resident MSCs (LMSCs) improves recovery wound repair of A549 cells after scratch wounding. A549 cells were seeded in duplicates in ECIS arrays and grown to confluence for 48 h. Medium was refreshed and replaced by control medium, medium containing 1% FCS (positive control) or conditioned medium (CM) of LMSCs from 6 non-COPD control and 6 COPD donors. After 24 h, cells were wounded by scratching. Relative wound closure was assessed by measuring the average length from one side of the scratch to the other at 0, 24 and 48 hrs derived from 3 measurements using ImageJ. (**A**) A representative light microscopy image of scratching of the A549 monolayer and its recovery. (**B**) Percentage wound closure in A549 cells treated with or without 1% FCS (positive control) or LMSC-CM from non-COPD control and COPD donors (*n* = 12, non-COPD and COPD LMSC-CM combined). Mean ± SEM levels are shown. (**C**) Percentage wound closure in A549 cells treated with LMSC-CM from non-COPD control or COPD donors (*n* = 6/group). Mean ± SEM levels are shown. To test for differences between control medium and LMSC-condition medium, 2-way ANOVA was used. * = *p* < 0.05 and ** = *p* < 0.01 between the indicated values.

**Figure 4 cells-10-02860-f004:**
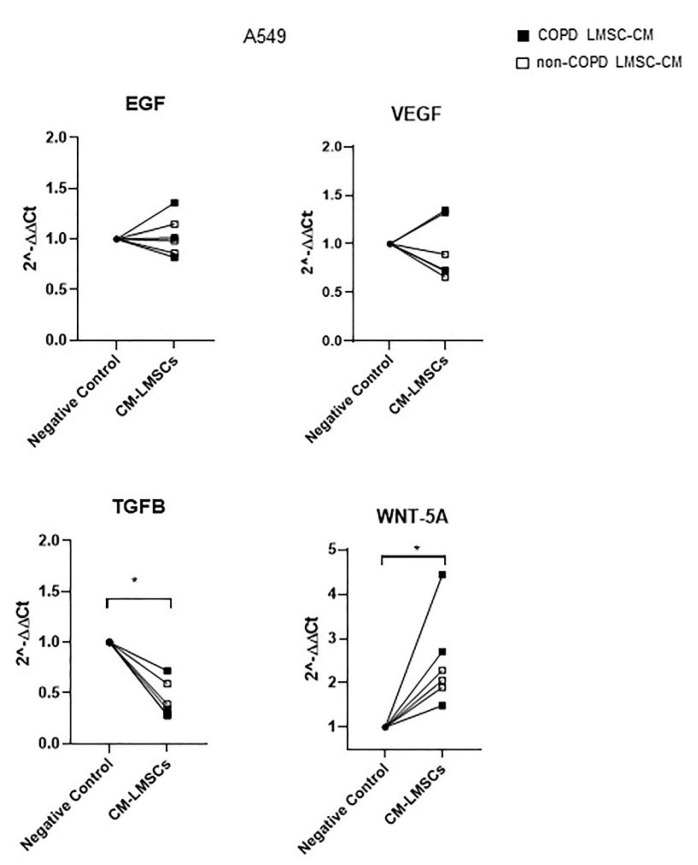
Conditioned medium from lung-resident MSCs (LMSCs) decreases the mRNA expression TGF-β and increases the mRNA expression of WNT-5A in A549 cells after scratch wounding. A549 cells were seeded in duplicates and grown to confluence for 48 h. Medium was refreshed and replaced by control medium or conditioned medium (CM) of LMSCs from 3 non-COPD control and 3 COPD donors. Cells were harvested for RNA isolation after 48 h and expression of *EGF*, *VEGF*, *TGFB* and *WNT5A* was related to the expression of the housekeeping genes *B2M* and *PPIA* and expressed as 2^−ΔΔCt^ compared to baseline. Data are presented as mean ± SEM. LMSC-CM from non-COPD donors is indicated by open symbols, from COPD donors by closed symbols. * = *p* < 0.05 between the indicated values as analyzed by the Wilcoxon signed rank test for paired observations.

**Figure 5 cells-10-02860-f005:**
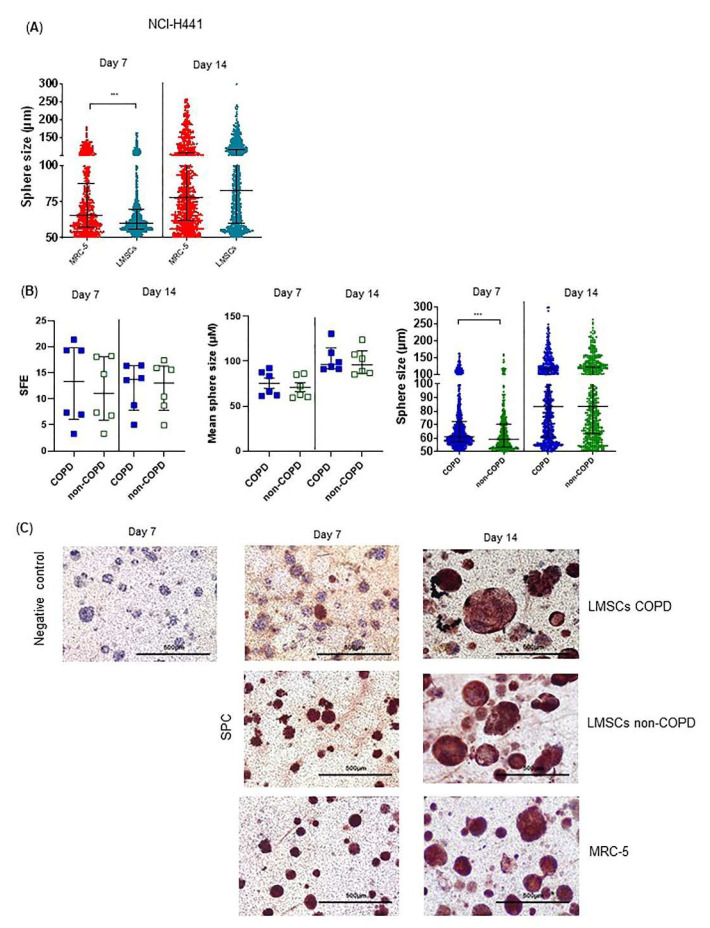
Abnormalities in COPD-derived lung-resident MSCs (LMSCs)-supported organoid formation by NCI-H441 cells. NCI-H441 cells together with mitomycin-treated MRC-5 cells or mitomycin-treated LMSCs from COPD and control donors were seeded into 100 μL growth factor-reduced 1:1 diluted Matrigel onto inserts of a transwell and cultured for 7–14 days. The number per well and the size of organoids were measured using light microscopy. The sphere forming efficiency (SFE) was determined as number of organoids normalized by cell input. (**A**) Size of organoids generated in the presence of LMSCs (*n* = 12, COPD and control donors combined) or MRC-5 cells. Medians ± IQR are indicated. (**B**) SFE, mean size and size distribution of organoids generated in the presence of COPD (*n* = 6) and non-COPD (*n* = 6) LMSCs. Medians ± IQR are indicated. (**C**) Inserts were stained with for SPC (using AEC and HE counterstaining). Representative images are shown of 2 independent experiments with 3 non-COPD and 3 COPD LMSC donors per experiment. *** = *p* < 0.001, * = *p* < 0.05 between the indicated values as analyzed by the Mann Whitney U test.

**Figure 6 cells-10-02860-f006:**
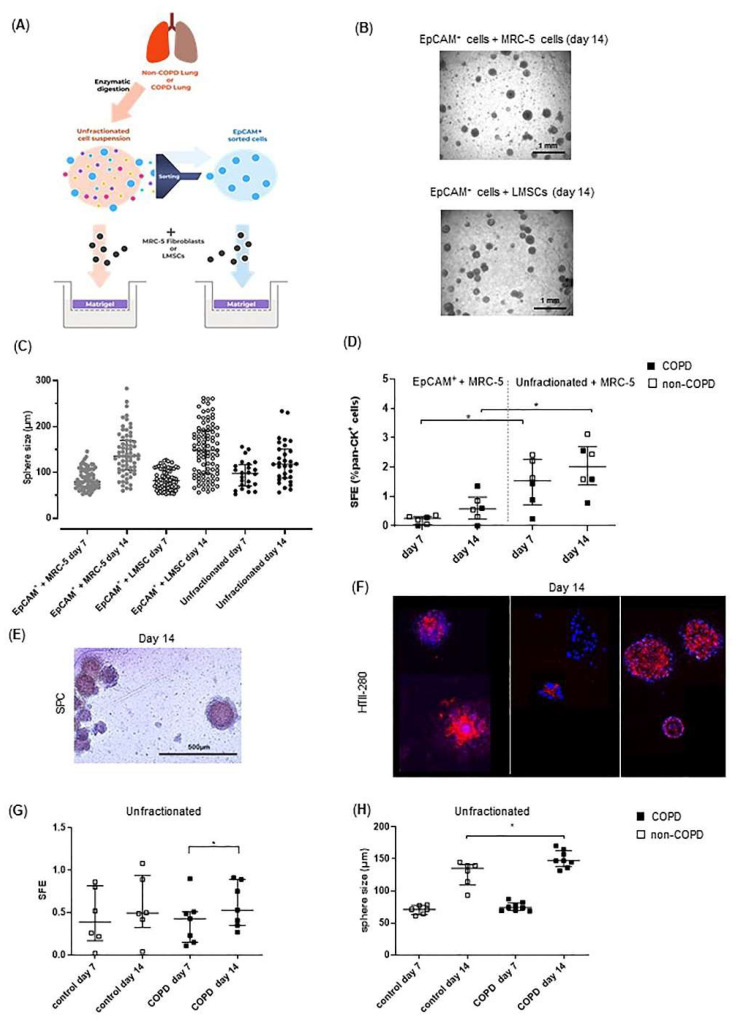
Lung-resident MSCs (LMSCs) support organoid formation by epithelial cells from human distal lungs and organized generated from COPD lung suspensions develop similar abnormalities. Primary human lung cells together with mitomycin-treated MRC-5 cells or mitomycin-treated LMSCs from COPD and control donors were seeded into 100 μL growth factor-reduced 1:1 diluted Matrigel onto inserts of a transwell and cultured for 7–14 days. The number per well and the size of organoids was measured using light microscopy. (**A**) Schematic picture of the primary lung cell isolation and organoid culture model. (**B**) Representative light microscopic images of organoids generated from EpCAM^+^ progenitors from human lungs generated with MRC-5 or LMSCs support. (**C**) Sphere forming efficiency (SFE) of organoids generated from EpCAM^+^ and unfractionated human lung cell suspensions (from a COPD donor) in absence or presence of MRC-5 cells or LMSCs (pooled fractions from 3 non-COPD donors) at day 7 and day 14. (**D**) Sphere forming efficiency (SFE) of organoids generated from EpCAM^+^ and unfractionated human lung cell suspensions from 3 COPD donors and 3 non-COPD donors at day 7 and day 14. (**E**) Inserts from a COPD donor at day 14 were stained for SPC (using AEC and HE counterstaining). (**F**) Inserts from 3 COPD donors at day 14 were stained for HTII-280 (using Alexa-Fluor 647-labeled antibody and DAPI to visualize nuclei). Representative images are shown. (**G**) Sphere forming efficiency (SFE) and (**H**) mean size of organoids generated from EpCAM^+^ and unfractionated human lung cell suspensions from 8 COPD donors and 6 non-COPD donors in presence of MRC-5 cells at day 7 and day 14. * = *p* < 0.05 between the indicated values as analyzed by the Mann–Whitney U test.

**Table 1 cells-10-02860-t001:** Characteristics of subjects included in the study.

	Non-COPD Control (*n* = 12)	COPD GOLD III–IV (*n* = 12)
Sex (M/F/NA)	3/7/2	3/9/0
Smoking (current/ex/never)	2/5/3	0/12/0
Age (years-range)	66.2 (53–79)	58.7 (51–68)
FEV1%Pred	106.4 (80–131)	25.0 (12–63)
FEV1/FVC	75.6 (65.0–89.9)	38.8 (21.9–89.2)

FEV1%Pred = Predicted value for Forced Expiratory Volume in 1 s; FEV1 = Forced Expiratory Volume in 1 s; FVC = Forced Vital Capacity. NA = not available. For age, FEV1%Pred and FEV1/FVC, group medians with ranges are shown. Exclusion criteria for subject inclusion in the study were the diagnosis of asthma, indications of lung infection, COPD GOLD stage classification of I or II or abnormalities in tissue structure.

## Data Availability

The data presented in this study are available upon request from the corresponding author.

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
