# Peer review of "Paracrine Regulation of Alveolar Epithelial Damage and Repair Responses by Human Lung-Resident Mesenchymal Stromal Cells"

_cells, 2021, doi:10.3390/cells10112860_

Round 1

Reviewer 1 Report

This paper entitled: “Paracrine regulation of alveolar epithelial damage and repair 2 responses by human lung-resident mesenchymal stromal cells” by Kruk et colleagues, resulted well written and clear.

However, some points can be improved:

  • Section 2.1: better specify the characteristics of included/excluded patients. What kind of tumor the patients are affected?
  • Are There a possibility to use as “control” group healthy subjects?
  • Can the authors also added other functional parameters (such as FVC, TLCO…?)?
  • “Spheres of a diameter >50 μm were defined as organized and counted.” Please add a reference for supporting this observation.
  • Improve the discussion section by adding recently published references in this field

Author Response

General comments: This paper entitled: “Paracrine regulation of alveolar epithelial damage and repair 2 responses by human lung-resident mesenchymal stromal cells” by Kruk et colleagues, resulted well written and clear.

Response: We thank the reviewer for the careful evaluation and the useful suggestions for our manuscript, and for considering it clear and well written.

Major comments:

Comment 1. Section 2.1: better specify the characteristics of included/excluded patients. What kind of tumor the patients are affected?

Response: The patient characteristics of the included subjects have been presented in Table 1. All patient data have been anonymized, without identifiable information being reported to the researchers. The information regarding the type of tumor was not available to the researchers. It may be relevant to note that the resections performed in our institute comprise those for metastatic cancers from outside the lung as well as smoking-unrelated lung tumors, like carcinoid and well-differentiated adenocarcinomas (the latter are the most frequent primary lung cancers in non-smokers). Importantly, tissue was collected as far distant from the tumor as possible, checked for abnormalities by an experienced pathologist and if indicated, the tissue was excluded from our study (see Methods section on page 3, lines 102-104).

Comment 2. Are There a possibility to use as “control” group healthy subjects?

Response: We agree with the reviewer that a healthy control group would be ideal. It is important to stress, however, that MSCs were isolated from distal lung tissue. This cannot be obtained from living donors without a surgical procedure, which would be unethical to perform in healthy subjects.

Comment 3. Can the authors also added other functional parameters (such as FVC, TLCO…?)?

Response: The FEV1%Pred and FEV1/FVC values have been provided in Table 1. FEV1/FVC is the standard parameter used by the Global Initiative for Chronic Obstructive Lung Disease (GOLD) to indicate the presence of airway obstruction. Unfortunately, TLCO data as measurement of gas exchange were not available for the majority of patients, as TLCO tests are not routinely performed.

Comment 4. “Spheres of a diameter >50 μm were defined as organized and counted.” Please add a reference for supporting this observation.

Response: We have now added the following reference, where the used protocol for organoid culture was established and optimized (Ng-Blichfeldt, J. P. et al. Retinoic acid signaling balances adult distal lung epithelial progenitor cell growth and differentiation. EBioMedicine 36, 461-474 (2018)). Furthermore, we have corrected the sentence, replacing “organized” by organoid.

Comment 5. Improve the discussion section by adding recently published references in this field.

Response: We thank the reviewer for the useful suggestion. To improve the discussion, we have now included the reference to a recent study on the supporting effect of murine lung MSCs (LMSCs) on alveolar organoid formation and defects herein in LMSCs derived from aged mice (Chanda, D. et al. Mesenchymal stromal cell aging impairs the self-organizing capacity of lung alveolar epithelial stem cells. Elife. 10:e68049 (2021)). This has been included on page 15, lines 476-478.

Reviewer 2 Report

I have some comments:

1 - Abstract. Co-culture with LMSCs supported organoid formation  in NCI-H441 and primary epithelial cells, resulting in larger organoids with lower type II-marker  positivity in the presence of COPD-derived versus control LMSCs. Similar abnormalities developed  in organoids from COPD compared to control-derived lung cells. Collectively, this indicates that  LMSCs’ secretome attenuates alveolar epithelial injury and supports epithelial repair. Additionally, LMSCs promote generation of alveolar organoids, with abnormalities in the supportive effects of  COPD-derived LMCS, reflective of impaired regenerative responses of COPD distal lung cells. This  identifies LMSCs not only as strategy but also target for future therapies. Could you please ameliorate the description of the aim of the study?

2- Abstract. Organoid number, size and expression of alveolar type II markers  were assessed. Pre-treatment with LMSC-CM significantly attenuated oxidative stress-induced necrosis and accelerated wound repair in A549. Co-culture with LMSCs supported organoid formation  in NCI-H441 and primary epithelial cells, resulting in larger organoids with lower type II-marker  positivity in the presence of COPD-derived versus control LMSCs. Please add the most important results and he statistical values to support the conclusions?

3 - 2. Materials and Methods. 2.1 Subjects. Distal lung tissue was derived from a total of 12 emphysema patients with GOLD stage III-IV COPD undergoing lung transplantation or lung volume reduction surgery and from leftover lung material of 12 non-COPD controls undergoing tumor resection surgery.
Do you have any comments regarding the limited number of patients in the two groups?

4 - 2.14 Statistics The Mann Whitney U test was used when testing for differences between two groups and  the Wilcoxon signed rank test was used for paired comparisons between conditions  within groups. The Friedman test with Dunn’s correction was used when multiple groups  and conditions were compared and organoid size. Two-way ANOVA was used to compare time elapse curves in the wounding assays. Please ameliorate this paragraph.

5 - 4. Discussion  In the current study, we investigated whether endogenous LMSCs exert beneficial effects  on distal lung epithelial damage and repair responses within the context of COPD. We observed that LMSCs are able to reduce damage in response to oxidative stress and to  promote repair in response to different types of injury in a paracrine manner. Additionally, LMSCs supported alveolar epithelial regenerative responses in an organoid model. Pretreatment of the alveolar epithelial cell line A549 with conditioned-medium from  LMSCs protected against H2O2-induced cell death, without a significant difference in the  effect of LMSCs from COPD and non-COPD lungs. Please underline the novelty of your research and better support the study results.6 - 4. Discussion. Please summarise this section. 7 - 5. Conclusions Collectively, we show that paracrine factors from LMSCs reduce distal lung epithelial damage and support repair and regenerative responses. LMSCs from COPD lungs show  specific abnormalities in the support of alveolar regenerative responses as demonstrated  by organoid models, which may reflect impaired differentiation into type II cells. These  abnormalities should be taken into account when considering autologous LMSCs for cell- based therapeutic strategies in COPD. Moreover, our study identifies endogenous LMSCs as target for therapeutic strategies, contributing to abnormal lung tissue repair in COPD. Please improve the conclusions.

Author Response

Major comments:

Comment 1. Abstract. Co-culture with LMSCs supported organoid formation  in NCI-H441 and primary epithelial cells, resulting in larger organoids with lower type II-marker  positivity in the presence of COPD-derived versus control LMSCs. Similar abnormalities developed  in organoids from COPD compared to control-derived lung cells. Collectively, this indicates that  LMSCs’ secretome attenuates alveolar epithelial injury and supports epithelial repair. Additionally, LMSCs promote generation of alveolar organoids, with abnormalities in the supportive effects of  COPD-derived LMCS, reflective of impaired regenerative responses of COPD distal lung cells. This  identifies LMSCs not only as strategy but also target for future therapies. Could you please ameliorate the description of the aim of the study?

Response: We hypothesized that lung-derived mesenchymal stromal cells (LMSCs) reduce alveolar epithelial damage and/or promote repair via paracrine processes, which may be impaired in COPD. We indeed observed abnormalities in the support of organoid formation by COPD-derived LMSCs. In our conclusion, we therefore mentioned that LMSCs may be a target in future therapeutic strategies. This would particularly be with respect to the production of specific growth factors. MSCs have been widely used in cell-based therapeutic strategies, although the effects of LMSCs have not been studied extensively, and the effectiveness of autologous COPD-derived LMSCs may be hampered. To improve the description of the aim, we have now adapted the hypothesis/aim as follows:

We hypothesized that lung-derived mesenchymal stromal cells (LMSCs) reduce alveolar epithelial damage and/or promote repair via paracrine processes, and may thus be suitable for cell-based strategies in COPD. We aimed to assess whether COPD-derived LMSCs display abnormalities which may be impaired in COPD”.

Because the abstract would now exceed the word limit, and to improve clarity, we have now removed “This  identifies LMSCs not only as strategy but also target for future therapies” from the conclusion of the abstract.

Comment 2. Abstract. Organoid number, size and expression of alveolar type II markers  were assessed. Pre-treatment with LMSC-CM significantly attenuated oxidative stress-induced necrosis and accelerated wound repair in A549. Co-culture with LMSCs supported organoid formation  in NCI-H441 and primary epithelial cells, resulting in larger organoids with lower type II-marker  positivity in the presence of COPD-derived versus control LMSCs. Please add the most important results and he statistical values to support the conclusions?

Response: Because of the word limit of 200, we have omitted the p values in the abstract. When adding these, we will then have to compromise on the description of the results, which we decided not to do because of clarity reasons. We have mentioned the most relevant findings in the abstract, and have now ensured that we always indicated this when differences were significant.  

Co-culture with LMSCs supported organoid formation in NCI-H441 and primary epithelial cells, resulting in significantly larger organoids with lower type II-marker positivity in the presence of COPD-derived versus control LMSCs. Similar abnormalities developed in organoids from COPD compared to control-derived lung cells, with significantly larger organoids”.

Comment 3. Section 2 -  Materials and Methods. 2.1 Subjects. Distal lung tissue was derived from a total of 12 emphysema patients with GOLD stage III-IV COPD undergoing lung transplantation or lung volume reduction surgery and from leftover lung material of 12 non-COPD controls undergoing tumor resection surgery. Do you have any comments regarding the limited number of patients in the two groups?

Response: For these type of studies, functionally assessing responses of patient-derived cells in vitro, a sample size of n=4-8 is usually sufficient to demonstrate differences between groups. Using cell lines, with less heterogeneity, this is usually even a lower number of n=3-6. Therefore, we are confident that the number of 12 subjects per group is sufficient and should not be regarded as limited.

Comment 4. Section 2.14 – Statistics. The Mann Whitney U test was used when testing for differences between two groups and  the Wilcoxon signed rank test was used for paired comparisons between conditions within groups. The Friedman test with Dunn’s correction was used when multiple groups  and conditions were compared and organoid size. Two-way ANOVA was used to compare time elapse curves in the wounding assays. Please ameliorate this paragraph.

Response: We are not fully sure which parts of this paragraph are unclear to the reviewer. We would like to stress that we have mentioned in the figure legends which test was used for which experiment, and we feel that it would be redundant to add this in the Methods section. However, we noticed two errors. We apologize and thank the reviewer for pointing this out. We have now corrected this as follows:

Page 6, lines 248-253:

“The Mann Whitney U test was used when testing for differences between two groups and the Wilcoxon signed rank test was used for paired comparisons between conditions within groups. The Friedman test with Dunn’s correction was used when multiple groups and conditions were compared and organoid size. Two-way ANOVA was used to compare time elapse curves in the wounding assays. P<0.05 was considered statistically significant”.

Comment 5. Section  4 – Discussion. In the current study, we investigated whether endogenous LMSCs exert beneficial effects on distal lung epithelial damage and repair responses within the context of COPD. We observed that LMSCs are able to reduce damage in response to oxidative stress and to  promote repair in response to different types of injury in a paracrine manner. Additionally, LMSCs supported alveolar epithelial regenerative responses in an organoid model. Pretreatment of the alveolar epithelial cell line A549 with conditioned-medium from  LMSCs protected against H2O2-induced cell death, without a significant difference in the  effect of LMSCs from COPD and non-COPD lungs.  Please underline the novelty of your research and better support the study results.

Response: As mentioned on page 15, lines 474-475, we are the first to demonstrate the protective and supportive effects of lung-derived MSCs on epithelial damage and repair responses in a human setting. Previous studies have either used MSCs from a different source or used murine LMSCs. We have now added here that not only LMSCs, but also epithelial cells were from a human source. In the first paragraph on page 15, we have described our findings in more detail, as follows: 

Pretreatment of the alveolar epithelial cell line A549 with conditioned-medium from LMSCs protected against H2O2-induced cell death, without a significant difference in the effect of LMSCs from COPD and non-COPD lungs. When applying electric field-induced injury or scratch-wounding, pre-treatment with soluble factors from LMSCs resulted in accelerated recovery of the alveolar epithelial monolayer, improving migratory and proliferative responses, which was accompanied by increased gene expression of WNT5A and decreased expression of TGFB. LMSC-derived condition medium impaired the establishment of cell-cell contacts as measured by electrical resistance, potentially leading to reduced contact inhibition of migratory and proliferative epithelial responses. Again, we did not observe significant differences in the ability of COPD and non-COPD-derived LMSCs to promote epithelial wound repair responses. Finally, when comparing the ability of COPD and non-COPD-control derived LMSCs to support the generation of alveolospheres in an organoid model, we observed abnormalities in co-cultures with COPD-derived LMSCs. Here, larger organoids developed compared to organoids formed in the presence of control LMSCs. Importantly, similar abnormalities were observed in organoids formed by primary distal lung suspensions from COPD patients, indicating a defect that may be reflective of failing repair responses in COPD lungs”.

We are willing to add more detail here, although it would be helpful when the reviewer could specifically indicate which sections would need better description.

Comment 6. Section 4 - Discussion. Please summarise this section.

Response: We thank the reviewer for the useful suggestion.  We have now added the following page 17, lines 584-588:

“Together, using this approach, we were able to show that lung-resident MSCs play a critical role in the attenuation of alveolar epithelial damage, in which COPD-derived LMSCs do not display persistent, intrinsic defects upon in vitro expansion in 2D culture. LMSCs also support alveolar epithelial regeneration, in which COPD-derived LMSCs do display intrinsic abnormalities.”

Comment 7. Conclusions. Collectively, we show that paracrine factors from LMSCs reduce distal lung epithelial damage and support repair and regenerative responses. LMSCs from COPD lungs show  specific abnormalities in the support of alveolar regenerative responses as demonstrated  by organoid models, which may reflect impaired differentiation into type II cells. These  abnormalities should be taken into account when considering autologous LMSCs for cell- based therapeutic strategies in COPD. Moreover, our study identifies endogenous LMSCs as target for therapeutic strategies, contributing to abnormal lung tissue repair in COPD. Please improve the conclusions.

Response: We have now adapted the conclusions as follows:

Collectively, we show that paracrine factors from LMSCs reduce distal lung epithelial damage and support repair and regenerative responses. LMSCs from COPD lungs show specific abnormalities in the support of alveolar regenerative responses as demonstrated by organoid models the formation of larger organoids. Larger organoids were also generated from lung cell suspensions of COPD patients versus non-COPD controls, and which may reflect impaired differentiation into type II cells. These abnormalities should be taken into account when considering autologous LMSCs for cell-based therapeutic strategies in COPD. Moreover, our study identifies endogenous LMSCs as target for therapeutic strategies, contributing to abnormal lung tissue repair in COPD.

Round 2

Reviewer 2 Report

The manuscript has been improve: I have only few suggestion:

1) 1. Introduction L35
Chronic Obstructive Pulmonary Disease (COPD) is a prevalent inflammatory lung disease that currently is the 3rd leading cause of death worldwide according to WHO (WHO, 2018). The main risk factor for COPD is the inhalation of noxious particles and gasses, such as cigarette smoke and air pollutants, which in combination with genetic susceptibility lead to inflammation, lung tissue damage and aberrant tissue repair in COPD patients. The disease is characterized by mucus hypersecretion (chronic bronchitis), airway wall thickening and/or destruction of the alveoli with airspace enlargement (emphysema), leading to airflow limitation. Currently, there is no cure for the disease and the alveolar damage is irreversible. Therefore, there is an urgent need for novel treatment strategies 
that halt or reverse the progressive loss of lung function. Could you please add some information regarding the alveolar damage and include these references:

a- Tuder RM, Petrache I. Pathogenesis of chronic obstructive pulmonary disease. J Clin Invest. 2012 Aug;122(8):2749-55. doi: 10.1172/JCI60324.

b- Ruaro B, Salton F, Braga L, Wade B, Confalonieri P, Volpe MC, Baratella E, Maiocchi S, Confalonieri M. The History and Mystery of Alveolar Epithelial Type II Cells: Focus on Their Physiologic and Pathologic Role in Lung. Int J Mol Sci. 2021 Mar 4;22(5):2566. doi: 10.3390/ijms22052566.

Author Response

Comment 1. Could you please add some information regarding the alveolar damage and include these references.

Response 1. We thank the reviewer for the additional useful suggestion to improve our manuscript. We have now added a reference regarding the alveolar damage in COPD. We included the following reference: Tuder RM, Petrache I. Pathogenesis of chronic obstructive pulmonary disease. J Clin Invest. 2012 Aug;122(8):2749-55, as suggested by the reviewer. We feel that of the two articles suggested by the reviewer, this article most adequately describes the alveolar damage in emphysema.